# Repetitive transcranial magnetic stimulation (rTMS) triggers dose-dependent homeostatic rewiring in recurrent neuronal networks

**Swathi Anil**[1,2,3], **Han Lu**[1,4], **Stefan Rotter**[2,3,4‡]*, **Andreas Vlachos**[1,2,4,5‡]*

**1** Department of Neuroanatomy, Institute of Anatomy and Cell Biology, Faculty of Medicine, University of Freiburg, Freiburg, Germany, **2** Bernstein Center Freiburg, University of Freiburg, Freiburg, Germany, **3** Faculty of Biology, University of Freiburg, Freiburg, Germany, **4** Center BrainLinks-BrainTools, University of Freiburg, Freiburg, Germany, **5** Center for Basics in NeuroModulation (NeuroModulBasics), Faculty of Medicine, University of Freiburg, Freiburg, Germany

‡ These authors are joint senior authors on this work.
\* stefan.rotter@bio.uni-freiburg.de (SR); andreas.vlachos@anat.uni-freiburg.de (AV)

**Data Availability Statement:** All code is provided in the Github repository https://github.com/swathianil/homeostatic_structural_plasticity_rTMS and data supporting the results of this study can be

## Abstract

Repetitive transcranial magnetic stimulation (rTMS) is a non-invasive brain stimulation technique used to induce neuronal plasticity in healthy individuals and patients. Designing effective and reproducible rTMS protocols poses a major challenge in the field as the underlying biomechanisms of long-term effects remain elusive. Current clinical protocol designs are often based on studies reporting rTMS-induced long-term potentiation or depression of synaptic transmission. Herein, we employed computational modeling to explore the effects of rTMS on long-term structural plasticity and changes in network connectivity. We simulated a recurrent neuronal network with homeostatic structural plasticity among excitatory neurons, and demonstrated that this mechanism was sensitive to specific parameters of the stimulation protocol (i.e., frequency, intensity, and duration of stimulation). Particularly, the feedback-inhibition initiated by network stimulation influenced the net stimulation outcome and hindered the rTMS-induced structural reorganization, highlighting the role of inhibitory networks. These findings suggest a novel mechanism for the lasting effects of rTMS, i.e., rTMS-induced homeostatic structural plasticity, and highlight the importance of network inhibition in careful protocol design, standardization, and optimization of stimulation.

## Author summary

The cellular and molecular mechanisms of clinically employed repetitive transcranial magnetic stimulation (rTMS) protocols remain not well understood. However, it is clear that stimulation outcomes depend heavily on protocol designs. Current protocol designs are mainly based on experimental studies that explored functional synaptic plasticity, such as long-term potentiation of excitatory neurotransmission. Using a computational approach, we sought to address the dose-dependent effects of rTMS on the structural remodeling of stimulated and non-stimulated connected networks. Our results suggest a new mechanism of action—activity-dependent homeostatic structural remodeling—

found in Zenodo at https://doi.org/10.5281/zenodo.8374484.

**Funding:** This work was supported by NIH (1RO1NS109498 to AV) and by the Federal Ministry of Education and Research, Germany (BMBF, 01GQ2205A to AV). We acknowledge the support by the state of Baden-Württemberg, Germany through bwHPC and the German Research Foundation (DFG) through grant no INST 39/963-1 FUGG (bwForCluster NEMO). The funders had no role in study design, data collection and analysis, decision to publish, or preparation of the manuscript.

**Competing interests:** The authors have declared that no competing interests exist.

through which rTMS may assert its lasting effects on neuronal networks. We showed that the effect of rTMS on structural plasticity critically depends on stimulation intensity, frequency, and duration and that recurrent inhibition can affect the outcome of rTMS-induced homeostatic structural plasticity. These findings emphasize the use of computational approaches for an optimized rTMS protocol design, which may support the development of more effective rTMS-based therapies.

## Introduction

Repetitive transcranial magnetic stimulation (rTMS) is a non-invasive brain stimulation method used in basic and clinical neuroscience [1–3]. Based on the principle of electromagnetic induction, rTMS induces electric fields that activate cortical neurons and modulate cortical excitability beyond the stimulation period [4–6]. This makes rTMS a suitable tool for studying and modulating brain plasticity in healthy and disease states [7–11].

Experiments in animal models have shown that rTMS induces specific changes in excitatory synapses, that are consistent with a long-term potentiation (LTP) of neurotransmission [12–15]. Using animal models (both *in vitro* and *in vivo*), we previously also demonstrated rTMS-induced changes in inhibitory neurotransmission, wherein a reduction in dendritic but not somatic inhibition was observed [16]. These findings provide an explanation of how rTMS may assert its effects—by mediating disinhibition and priming stimulated networks for the expression of physiological context-specific plasticity [17]. Nevertheless, it remains unknown how exogenous electric brain stimulation that is not linked with specific environmental or endogenous signals asserts therapeutic effects in patients.

In recent years, a considerable degree of variability (or even absence) of rTMS induced "LTP-like" plasticity—measured as a change in the evoked potential of the target muscle upon stimulation of the motor cortex [18–21]—has been reported in human participants, often leading to difficulties in reproducing results [22]. Efforts to explain this variability have largely focused on the assessment of possible confounding factors that may affect the outcome of a given rTMS protocol as well as on prospective optimization of induced electrical fields for standardization of stimulation protocols and dosing across participants [23, 24]. This has also led to discussions on alternative underlying mechanisms, such as the impact of rTMS on glial cells and rTMS-induced structural remodeling of neuronal networks [25–28]. There has been emerging evidence of structural plasticity induced by rTMS. Studies have demonstrated that rTMS facilitates the reorganization of abnormal cortical circuits [10, 11], which may be pertinent to its therapeutic effects and cognitive benefits [29, 30]. Moreover, structural connectivity changes induced by rTMS have been shown to underlie anti-depressant effects in chronic treatment-resistant depression [31–33]. Vlachos et al.[12] also demonstrated structural remodeling imposed by 10 Hz repetitive magnetic stimulation on small dendritic spines in an *in vitro* setting. More recently, structural synaptic plasticity in response to low-intensity rTMS was demonstrated using longitudinal two-photon microscopy in the motor cortex of mice [14]. Towards this direction, we used network simulations to evaluate the dose-dependent effects of rTMS on the structural remodeling of neuronal networks in this study. We employed an inhibition-dominated, sparsely connected recurrent neuronal network model using leaky integrate-and-fire point neurons to capture network dynamics influenced by rTMS. The model integrates three critical components of cortical networks: asynchronous irregular neuronal activity, inhibition domination, and sparse connectivity. Asynchronous irregular neuronal activity, a distinctive feature of healthy cortical

networks, is essential for realistic representation of cortical network dynamics under rTMS. The dynamics of individual neurons is characterized by stochastic firing, diverse input integration, absence of intrinsic oscillations, and sensitivity to input. Inhibition domination, which underscores the significant role of inhibitory connections in network activity and balance, was included in the model to accurately capture the inhibitory effects on the network dynamics under the influence of rTMS. In an inhibition-dominated network, inhibitory signals help to prevent the network from becoming overly synchronized, which can promote asynchronous firing. Sparse connectivity, a fundamental property of cortical networks where only a small fraction of neurons are interconnected, was incorporated to reflect a realistic cortical network architecture. We opted for a simplistic leaky integrate-and-fire neuronal model, which provided a good balance in terms of computational feasibility while still capturing essential neuronal dynamics relevant to our study. Leaky integrate-and-fire neurons effectively encapsulate asynchronous irregular firing. This ability to replicate irregular and independent firing patterns is particularly crucial to our study, as it involves homeostatic plasticity mechanisms which are likely to induce changes in firing patterns. We evaluated rTMS-induced structural changes that may occur even in the absence of changes in synaptic weights (i.e., LTP-like plasticity). Specifically, we employed homeostatic structural plasticity which follows a negative feedback rule [34–36] in our network. In this network, continuous synaptic remodeling takes place in order to maintain neuronal activity at a stable level. Deviation from this level of activity are restored using synaptic formation or deletion at regular intervals. This rule has been previously demonstrated to have emergent associative properties [35]. This study showed that the  homeostatic structural plasticity rule led to the formation of a cell assembly among neurons that receive external stimulation, in absence of explicit correlation-based synaptic plasticity rules. They also showed the emergence of feature-specific connectivity as a result of sensory experience, similar to observations in the V1 region of mice. Additionally, network reorganisation caused by transcranial Direct Current Stimulation, another non-invasive brain stimulation technique, has been demonstrated using homeostatic structural plasticity [36]. This evidence supports the suitability of the homeostatic structural plasticity rule for capturing the changes in network connectivity induced by rTMS in our model. Moreover, the network characteristics we chose are commonly found across the neocortex in awake animals. Several studies have demonstrated the effect of rTMS in in vivo animal studies. Ma et al [29] were able to demonstrate that low-frequency rTMS plays an important role in the regulation of cognition-driven behavior by altering the synaptic structure of networks. Later, Ma et al [30] have shown that high-frequency rTMS can alleviate cognitive impairment and modulate hippocampal synaptic structural plasticity in aging mice. Tang et al [14] showed that rTMS can drive structural synaptic plasticity in the motor cortex of young and aging mice. Based on our previous experimental findings that 10 Hz stimulation induces structural remodeling of excitatory synapses and dendritic spines [12], we assessed the effects of stimulation intensity, pulse number, and frequency—including clinically established intermittent theta burst stimulation (iTBS)—on rTMS-induced homeostatic structural plasticity.

## Materials and methods

### Neuron model

All large-scale simulations in the present study were performed using NEST simulator 2.20.0 [37], using MPI-based parallel computation. Single neurons were modeled as linear current based leaky integrate-and-fire point neurons, having subthreshold dynamics expressed by the

**Table 1. Parameters of neuron model.**

| Parameter | Symbol | Value |
|---|---|---|
| Membrane time constant | $\tau_m$ | 20 ms |
| Resting potential | $V_{rest}$ | −60 mV |
| Threshold potential | $V_{th}$ | −40 mV |
| Excitatory postsynaptic potential (EPSP) amplitude | $J_E$ | 0.1 mV |
| Inhibitory postsynaptic potential (IPSP) amplitude | $J_I$ | −0.8 mV |
| Synaptic delay | $d$ | 2 ms |
| Reset potential | $V_{reset}$ | −50 mV |
| Refractory period | $t_{ref}$ | 2 ms |

following ordinary differential equation:

$$\tau_m \frac{dV_i}{dt} = -V_i + \tau_m \sum_j J_{ij} S_j(t-d) + \Delta V_{rTMS}, \tag{1}$$

where $\tau_m$ is the membrane time constant. The membrane potential of neuron $i$ is denoted by $V_i$. The neurons rest at −60 mV and have a firing threshold $V_{th}$ of −40 mV. The spike train generated by neuron $i$ is given by $S_i(t) = \sum_k \delta(t - t_i^k)$, where $t_i^k$ gives the individual spike times. The transmission delay is denoted by $d$. Individual excitatory postsynaptic potentials have the amplitude $J_E = 0.1$ mV, and inhibitory postsynaptic potentials have the amplitude $J_I = -0.8$ mV. The matrix entry $J_{ij}$ represents the amplitude of a postsynaptic potential induced in neuron $i$ when a spike from neuron $j$ arrives. As multiple synapses can exist from neuron $j$ to neuron $i$, the amplitude $J_{ij}$ is an integer multiple of $J_E$ or $J_I$, respectively, depending on the type of the presynaptic neuron. $\Delta V_{rTMS}$ denotes the membrane potential deviation induced by magnetic stimulation which will be introduced in the following section. An action potential is generated when the membrane potential $V_i(t)$ of the neuron reaches $V_{th}$, following which the membrane potential is reset to $V_{reset} = -50$ mV. All parameters are listed in Table 1.

## Network model

We implemented a sparsely connected recurrent neuronal network [38] comprising leaky integrate-and-fire point neurons, where inhibition dominated the dynamics (consisting of 10000 excitatory and 2500 inhibitory neurons, inhibitory synapses were stronger than excitatory ones). We did not intend to simulate a specific cortical region; instead, we integrated characteristics such as sparse connectivity, inhibition-domination, and asynchronous irregular firing among neurons, which are commonly observed across the neocortex in awake animals. Asynchronous irregular neuronal activity, known for its role for recurrent network activity [38–40], was faithfully reproduced in our cortical network model, consistent with previous studies [35, 36]. While more biophysically detailed models could have been employed, we opted for the leaky integrate-and-fire model to balance between computational feasibility and capturing essential neuronal dynamics relevant to our study. This model allowed us to systematically investigate the effects of rTMS on network dynamics and connectivity with the help of large-scale numerical simulations.

**Static network.**   All inhibitory synapses in the static network have a fixed synaptic amplitude of $J_I = -0.8$ mV and excitatory synapses have a fixed amplitude of $J_E = 0.1$ mV. All synapses among inhibitory neurons, excitatory neurons, and between excitatory and inhibitory neurons are static. These synapses are randomly established with a connection probability of

**Table 2. Parameters of static and plastic network models.**

| Parameter | Symbol | Value |
|---|---|---|
| Number of excitatory neurons | $N_E$ | 10000 |
| Number of inhibitory neurons | $N_I$ | 2500 |
| Connection probability | $C_p$ | 10% |
| Rate of external input | $r_{ext}$ | 30 kHz |

10%. All the neurons in the network receive steady stochastic background input in the form of Poisson spike trains of $r_{ext}$ = 30 kHz. This allows the neurons to have fluctuating subthreshold membrane potential dynamics with pre-determined stable firing rate of 7.8 Hz. The network parameters have been chosen to facilitate an asynchronous-irregular resting state. The network parameters have been listed in Table 2.

**Plastic network.**   The plastic network has the same network architecture as the static network, except that the E-E connections were grown from zero following the homeostatic structural plasticity rule implemented in previous works [35, 36, 41]. By setting the target firing rate to 7.8 Hz, the network will grow into an equilibrium state driven by the external Poisson input ($r_{ext}$ = 30 kHz), where the average connection probability is around 10% and all neurons fire irregularly and asynchronously around the target rate (7.8 Hz). While using a plastic network, any repetitive magnetic stimulation is only applied after the completion of the growth period. Network parameters can be found in Table 2.

## Homeostatic structural plasticity rule

As mentioned above, the connections among excitatory neurons (E-E) followed a homeostatic structural plasticity rule, and were subject to continuous remodeling. This rule has been inspired by precursor models by Dammasch [42], van Ooyen & van Pelt [43] and van Ooyen [44]. This specific model was previously employed to show cortical reorganization after stroke [45] and lesion [46], emergent properties of developing neural networks [47] and neurogenesis in adult dentate gyrus [48, 49]. However, we use a more recent implementation of this model in NEST [50] which does not include a distance-dependent kernel, previously used to demonstrate associative properties of homeostatic structural plasticity [35, 41]. The authors demonstrated that without the need for an enforced Hebbian plasticity rule, this homeostatic rule can cause network remodeling which displays emergent properties of Hebbian plasticity. Following external stimulation, the affected neurons underwent synaptic remodeling that lead to formation of a cell assembly among these neurons, thus exhibiting activity driven associativity, a distinctive feature of Hebbian plasticity [51]. In the present study, we follow this line of thought to propose an alternative mechanism of rTMS induced plasticity.

Each neuron $i$ in this model has a number of available axonal boutons (presynaptic elements, $z_i^{pre}$) and dendritic spines (postsynaptic elements, $z_i^{post}$), which are paired to form functional synapses. Synapses can only be formed if free synaptic elements are available. Each synapse has a uniform strength of $J_E$ = 0.1 mV. The growth rule we use is a rate-based rule, as implemented in NEST [50]. The rule follows the set-point hypothesis, which states that there is a set-point of intracellular calcium concentration that a neuron tries to achieve, in order to maintain stability. Deviations from this set-point level are met by global (whole neuron) efforts to restore it via synaptic turnover. This is in line with experimental results that have shown that neurite growth and deletion are controlled by intracellular calcium concentration [52–54]. Therefore, in the model of homeostatic structural plasticity used here, the growth and deletion of synaptic elements of a neuron $i$ are governed by its intracellular calcium

**Table 3. Parameters of structural plasticity model.**

| Parameter | Symbol | Value |
|---|---|---|
| Growth rate | $v$ | 0.0039 s$^{-1}$ |
| Target level of calcium | $\epsilon$ | 0.0078 |
| Time constant for calcium trace | $\tau_{Ca}$ | 10 s |
| Increment on calcium trace per spike | $\beta_{Ca}$ | 0.0001 |

concentration ($\phi_i(t) = [Ca^{2+}]_i$). Following each neuronal spike, there is an increase in intracellular calcium concentration by $\beta_{Ca}$ through calcium influx. The intracellular calcium concentration decays exponentially with time constant $\tau_{Ca}$ between spikes. The spike train $S_i(t)$ related intracellular calcium dynamics can be expressed as,

$$\frac{d\phi_i(t)}{dt} = -\frac{1}{\tau_{Ca}}\phi_i(t) + \beta_{Ca}S_i(t). \tag{2}$$

The variable $\phi_i(t)$ has been shown to be a good indicator of a neuron's firing rate [55]. According to the synaptic growth rule we use, each neuron $i$ maintains a time-varying estimate of its own firing rate, using its intracellular calcium concentration as a surrogate. This estimate is used by the neuron to control the number of its synaptic elements. When the firing rate falls below the prescribed set-point, indicated by a target firing rate, the neuron grows new synaptic elements to form additional synapses. Following this, freely available pre- and postsynaptic elements are randomly paired with free synaptic elements of other neurons, forming new synapses. These synapses enable the neuron to receive additional excitatory inputs, thus bringing the firing rate back to the set-point. Similarly, when the firing rate rises above the set-point, the neuron breaks existing synapses in order to limit the net excitatory inputs received. The elements from these broken synapses are added to the pool of free synaptic elements. Both the pre- and post-synaptic elements follow this linear growth rule [35, 36],

$$\frac{dz_i^k(t)}{dt} = v\left[1 - \frac{1}{\epsilon}\phi_i(t)\right], k \in \{\text{pre}, \text{post}\}, \tag{3}$$

where $i$ is the index of the neuron, $v$ is the growth rate and $\epsilon$ is the target level of calcium. The parameters of the homeostatic structural plasticity rule are listed again in Table 3.

## Model of repetitive transcranial magnetic stimulation (rTMS)

The electrical field induced by rTMS was implemented in the form of current injections into point neurons via a step-current generator in the NEST simulator. For mathematical simplification, TMS pulses were modeled as rectangular waves. Each stimulus pulse had a duration of 0.5 ms, inspired by the output of conventional rTMS devices, and was depolarizing (monophasic) in nature. Following evidence that rTMS causes changes in spiking activity of cortical pyramidal neurons [56–58], we used stimulation intensities that are suprathreshold in nature. This premise allowed us to simplify the role of TMS-induced electrical field in neuronal depolarisation in our simulations. The orientation of the e-field is known to influence the site of depolarisation in neurons, but since we use spatially simplistic point neurons, the site of stimulation does not have a specific influence as long as each stimulus causes an action potential.

In order to investigate the impact of protocol design, we modeled repetitive stimulation protocols (Fig 1D) of different frequencies and intensities. We also modeled the clinically relevant iTBS with 600 pulses, described in the following sections. This protocol has been

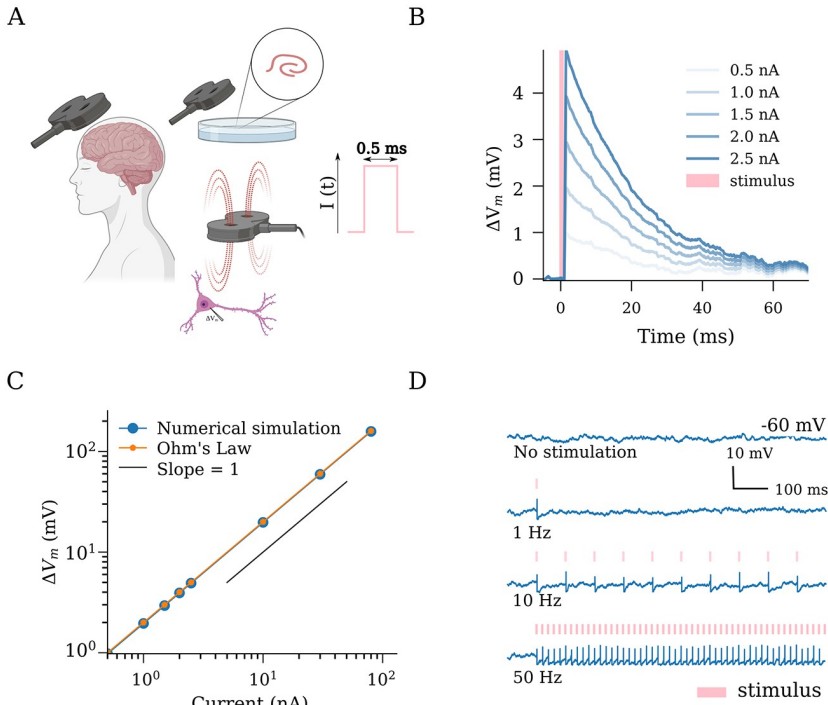

**Fig 1. Transcranial magnetic stimulation (TMS) has an immediate effect on the membrane potential dynamics of single neurons.** (A) Schematic illustration of TMS in humans and neurons. The TMS-induced electric fields cause depolarization of neurons in the target region. We implemented TMS as rectangular pulse current injections with a duration of 0.5 ms (c.f., output parameters of conventional TMS devices). (B) Single stimuli produce changes in the membrane potential in a dose-dependent manner. (C) A linear relationship is evident between applied effective stimulation strength (in nA) and the resulting membrane potential deviation, as predicted by Ohm's law (See Methods). (D) Suprathreshold stimulation at different frequencies elicits spiking responses from the stimulated neurons. Created with BioRender.com.

approved by the US Food and Drug Administration (FDA) for treating pharmacoresistant Major Depressive Disorder (MDD) [59]. Parameters of TMS protocols used throughout this study are summarised in Table 4.

## Numerical experimental protocols

**rTMS pulse-triggered membrane potential deviation.** In order to closely observe the response of individual neurons to single rTMS-like stimulus, we modeled single excitatory neurons that receive equal net excitatory and inhibitory Poisson inputs and therefore maintain subthreshold membrane potential dynamics. Spiking activity was disabled in the neuron. A single pulse current injection of 0.5 ms duration, which represents a magnetic stimulation pulse in our study, was delivered to the neurons. We observe the membrane potential trace 5 ms before the pulse onset to about 70 ms post the pulse onset. In order to account for randomness and variability, we traced membrane potentials of 500 isolated neurons, each receiving statistically independent Poisson inputs of the same rate. The membrane potential traces were averaged to obtain a robust readout. We repeat this experiment for different pulse amplitudes.

**Theta burst stimulation protocol.** Theta burst stimulation delivers bursts of stimuli at a 5 Hz frequency [60]. Each burst consists of three pulses that occur at a 50 Hz frequency. The US FDA approved iTBS protocol has a more temporally complex structure. The protocol consists

**Table 4. Parameters of Transcranial Magnetic Stimulation (TMS).**

| Fig | Protocol | Frequency (Hz) | $\Delta V_{rTMS}$(mV) | Pulse count | E-E synapses |
|---|---|---|---|---|---|
| 1B | single TMS | - | multiple[1] | 1 | - |
| 2C | rTMS | 1, 10, 50 | 0–200 | 900 | static |
| 2D | rTMS | 10 | multiple[2] | 100 | static |
| 2E | rTMS | 10, 20, 30, 40, 50 | 20–140 | 900 | static |
| 3C | rTMS | 10 | 68 | 900 | plastic |
| 3D | rTMS | 10 | multiple[2] | 900 | plastic |
| 4B | rTMS | 10 | 68 | multiple[3] | plastic |
| 4C | rTMS | 20, 30, 40, 50 | 68 | 300–3000 | plastic |
| 4D | rTMS | 5, 10, 15, 20 | multiple[2] | 600 | plastic |
| 5B | rTMS | iTBS | multiple[2] | 600 | plastic |
| 5C | rTMS | iTBS | 68 | multiple[4] | plastic |
| 5D | rTMS | iTBS, cTBS, 10 | 68 | multiple[4] | plastic |

[1] The membrane depolarisation caused are 0.98, 1.96, 2.94, 3.93, 4.92 mV.

[2] The membrane depolarization (mV) applied are $a = 20$, $b = 39$, $c = 68$, $d = 160$.

[3] The pulse numbers used are 900, 3000, 9000, and 22500.

[4] The pulse numbers used are 300, 600, 900, 1200, 3000, and 9000.

of 600 pulses that last a total duration of 192 s. The pulses are delivered in the theta burst format for 2 s, followed by an 8 s pause. This cycle is repeated 20 times. The continuous theta burst stimulation (cTBS) consists of 600 pulses in the theta burst format delivered in 40 s.

## Analysis and quantification

**Estimation of membrane potential deviation using Ohm's Law.**   The membrane potential deviation in the leaky integrate-and-fire neurons caused by a rTMS pulse, modeled as a current injection, was estimated using Ohm's Law. Accordingly, a current pulse of amplitude $A$ yields a membrane potential response, $\Delta V_{rTMS}$:

$$\Delta V_{rTMS} = AR\left[1 - \exp(-t/\tau_m)\right], \tag{4}$$

where $R = 80$ MΩ is the membrane leak resistance, $\tau_m = 20$ ms is the membrane time constant of the neuron. In the case of brief pulses, similar to the TMS pulses used in this study, following the current onset, the time course $\Delta V_{rTMS}$ of the voltage rises approximately linearly with time:

$$\Delta V_{rTMS} \approx AR\frac{t}{\tau_m}, \tag{5}$$

where $t = 0.5$ ms is the duration of the TMS pulse. We used the above equation to calculate the membrane potential deviation caused by TMS pulses to single neurons.

**Firing rate.**   The spiking activity of individual neurons is recorded using a spike detector available in NEST. The firing rate is then determined based on the average spike count within 1000 ms intervals, as a spike count average. The average firing rate for a population is determined by calculating the arithmetic mean of the individual firing rates of all neurons within that population.

**Network connectivity.**   Connectivity among all or subgroups of excitatory neurons is calculated using an $n{\times}m$ connectivity matrix $A_{ij}$, where n and m represents the total number of presynaptic and postsynaptic neurons, respectively. Each entry in this matrix can either be

zero or non-zero positive integers, denoting the total number of synapses from presynaptic neuron j to the postsynaptic neuron i. The connectivity of the whole network or subnetworks was used in the present study for any given time-point t. It is thus calculated as the mean number of synapses between two neurons, as follows:

$$C(t) = \frac{1}{nm}\sum_{ij} A_{ij}. \tag{6}$$

**Time constant of connectivity saturation.** In order to characterise the stimulation duration required to reach connectivity saturation during stimulation, we perform a curve-fitting of the data points using an exponential function $f(t) = ae^{-bt} + c$, where $\tau_{\text{decay}} = 1/b$ represents the time constant of the decay of connectivity during stimulation.

## Results

### Changes in single-neuron membrane potential dynamics and action potential induction in response to transcranial magnetic stimulation (TMS)-like electric stimulation

Multi-scale compartmental modeling demonstrates that the electric fields induced by TMS generally cause changes in the membrane potential of individual principal neurons, eventually resulting in action potential induction and characteristic intracellular calcium level changes [61–63]. Therefore, we first evaluated the effects of TMS-like electric stimulation on the membrane potentials at a single neuron level (Fig 1). For this purpose, single neurons—those receiving balanced excitatory and inhibitory Poisson spike trains—were stimulated with 0.5 ms rectangular current pulse injections of different amplitudes (Fig 1A and 1B). A linear interrelation between current injections and membrane potential deviation was observed, consistent with Ohm's law (Fig 1C). With this approach, implementation of suprathreshold repetitive stimulations, i.e., $\Delta V_{rTMS}$ = 68 mV at 1, 10 or 50 Hz, induced robust action potentials in the individual neurons (Fig 1D). We conclude that TMS-like neuronal spiking can be readily induced in our simulations.

### Non-linear effects of rTMS intensity on network activity

In realistic applications, TMS activates a network of connected neurons rather than a single neuron. Therefore, we evaluated the effects of increasing stimulation intensities on a subpopulation of neurons embedded in a recurrent network of 10000 excitatory and 2500 inhibitory neurons (Fig 2A). We modeled a focal stimulation that directly affected 10% of the excitatory neurons and studied the network response in terms of the firing rate changes among the following populations: stimulated excitatory neurons (S), non-stimulated excitatory neurons (E), and inhibitory interneurons (I). We first delivered a sample train of rTMS pulses (900 pulses at 10 Hz, c.f., [12, 64], with a pulse intensity that would cause a 68 mV membrane potential deviation) to the subpopulation. As shown in the raster plot, the spiking activity in the stimulated subpopulation was elevated (Fig 2B). We also observed a weaker synchronization throughout the subpopulations during stimulation, indicative of recurrent connectivity. Once stimulation ended, the neurons returned to their baseline Poisson firing patterns.

To examine the impact of different stimulation protocols on network activity, we performed a series of simulations with varying intensities and frequencies (each at 900 pulses). Examples of the firing rates of the defined subpopulations of interest are shown in Fig 2C.

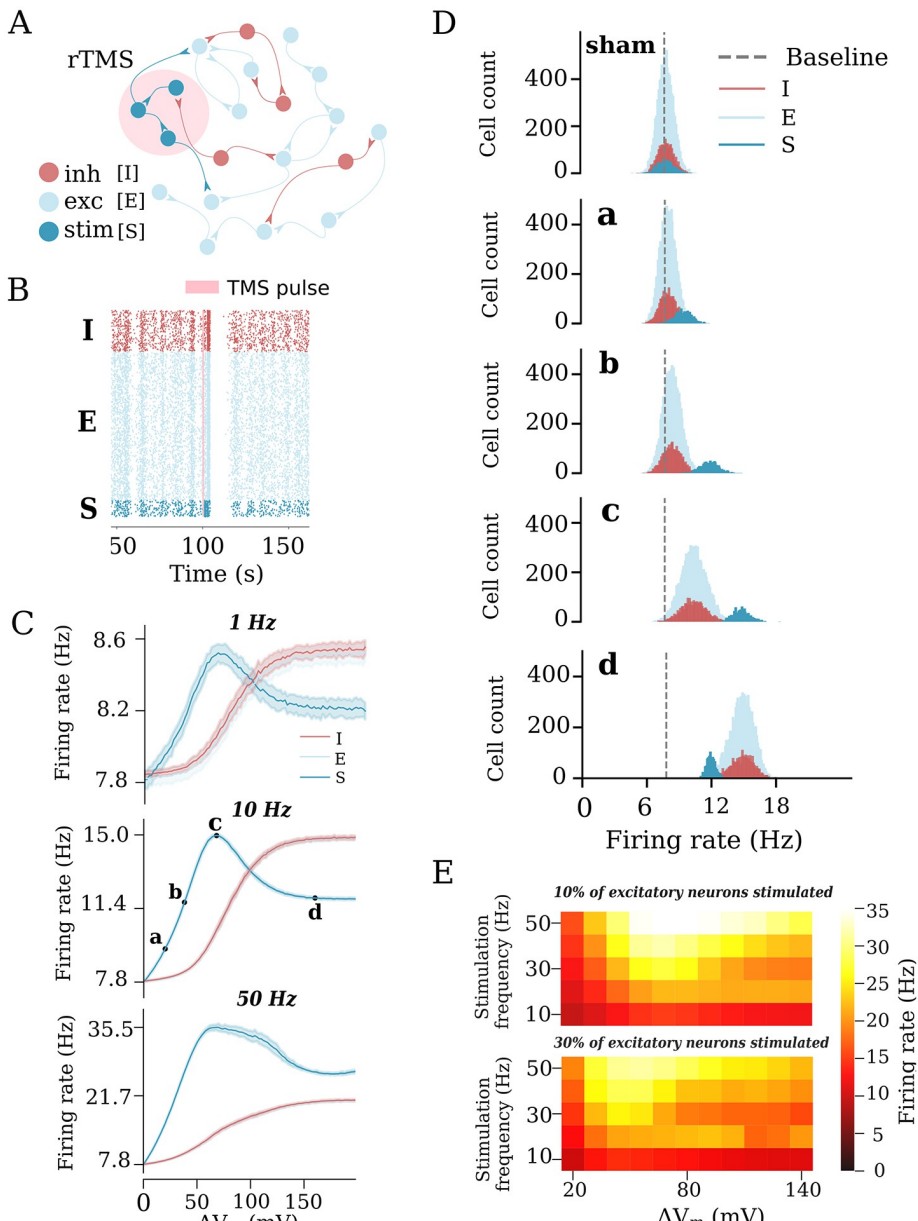

**Fig 2. Repetitive transcranial magnetic stimulation (rTMS) changes network activity in a static network.** (A) Illustration of the recurrent neuronal network with sparsely connected excitatory [E] and inhibitory [I] neurons used in this study. A subset of excitatory neurons [S] is stimulated. (B) rTMS influences the firing state of the stimulated neurons [S], causing them to fire in a more synchronous manner. (C) Change in the average firing rate in response to different stimulation intensities and frequencies of 10% of excitatory neurons. Four intensities (a: weak, c: peak, d: strong, and b: strong-equivalent) were selected to represent different stimulation intensities. (D) Firing rate histograms for populations E, I, and S at stimulation intensities a, b, c, and d, respectively. (E) Heatmaps summarizing the response of stimulated neurons to rTMS applied to 10% (top) and 30% (bottom) of excitatory neurons.

We found that the stimulated population responded at lower stimulation intensities and frequencies (i.e., 1 Hz and 10 Hz), with a proportional increase in the firing rates, which peaked at a stimulus-induced depolarization of 68 mV. With stronger stimulation, the firing rate response of the stimulated subpopulation declined as the firing rate of the inhibitory

neurons increased owing to recurrent inhibition. Eventually, a plateau was reached. For higher frequencies (i.e., 50 Hz), changes in the firing rate did not follow the exact same trend as for the lower frequencies (e.g., 1 Hz). This may be attributed to the strong high-frequency stimulation that forced the network to enter into a different stable regime. Nevertheless, the impact of recurrent inhibition on the stimulated neurons was still observable (Fig 2C).

The effects of different stimulation intensities on the network firing dynamics were carefully examined by plotting the firing rate distributions of the respective sub-populations in response to those intensities (Fig 2D). Weak stimulation did not cause noticeable additional activation of the inhibitory subpopulation. At the peak intensity, the inhibitory neurons were evidently activated. The strong stimulation significantly activated the inhibitory interneurons. The evoked recurrent inhibition had a profound effect on the stimulated subpopulation, resulting in suppression of its firing rate response. The same firing rate of the stimulated neurons was achieved at much lower stimulation intensities that did not recruit inhibition, including strong-equivalent intensity (c.f., Fig 2C). Based on these results, we selected four intensities, characteristic of different states of the network, for further exploration. The resulting values were expressed in terms of the induced changes in the membrane potential of the stimulated neurons and categorized as follows: (a) weak, 20 mV, (c) peak 68 mV, (d) strong, 160 mV and (b) strong-equivalent, 38 mV stimulations.

The results across a wide range of frequencies (10 to 50 Hz) and different stimulation intensities (20 to 140 mV-induced membrane potential change) are summarized in Fig 2E. The described effects on the inhibitory neurons and recurrent inhibition did not depend on the stimulation frequency. We also replicated these results in simulations of a larger subset of excitatory neurons (i.e., when 30% of the principal neurons were stimulated, Fig 2E, bottom). Herein, we observed lower peak firing rates of the stimulated neurons, demonstrating that recurrent inhibition was more effectively recruited when larger populations of neurons were directly stimulated. Taken together, these simulations suggest that an "optimal" stimulation intensity that effectively increases the firing rate of stimulated neurons exists. Exceeding this intensity leads to further recruitment of inhibition, which dampens the activity of the stimulated excitatory neurons. Lower strong-equivalent stimulation intensities can be determined at which the same effects on the firing rates of stimulated neurons are observed, without major effects on network inhibition.

## Structural remodeling of network connectivity in response to rTMS

We switched to a plastic network that remodels its connections in an activity-dependent homeostatic manner (Fig 3). This network follows a plasticity rule where an increase in the firing rate of excitatory neurons leads to retraction and disconnection, while a reduction in the firing rate promotes outgrowth and formation of new excitatory contacts between principal neurons (Fig 3A; c.f., [35, 36]). In this study, stimulation was performed after an initial growth stage, which allowed the network to reach a steady state with 10% connectivity between the excitatory neurons and a mean firing rate of 7.8 Hz (Fig 3B). We applied a 10 Hz stimulation protocol consisting of 900 pulses at peak intensity to a subset of 10% of excitatory neurons (c.f., Fig 2B). As described above, the stimulation elicited an instant increase in the firing rates of the stimulated neurons as well as non-stimulated excitatory and inhibitory neurons (Fig 3C). This sudden increase in the firing rates was accompanied with a homeostatic structural response where the principal neurons reduced existing input synapses to restore baseline activity. This disconnection was most prominently observed among the stimulated neurons, but also occurred between the stimulated and non-stimulated excitatory neurons (Fig 3C). The

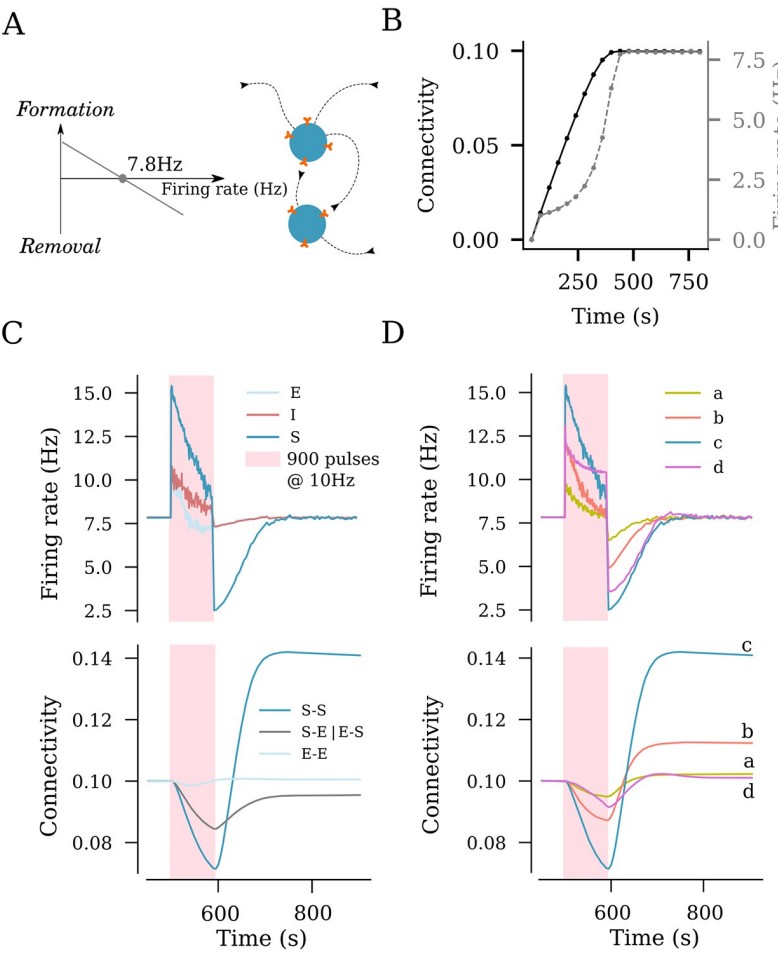

**Fig 3. rTMS induces structural remodeling of stimulated networks.** (A) Homeostatic structural plasticity assumes negative feedback of neuronal activity on its connectivity with other neurons: A high firing rate removes synapses between excitatory neurons, and a low firing rate promotes synapse formation. (B) Poisson input stabilizes the firing rate and connection probability prior to stimulation. (C) Effects of a 10 Hz stimulation protocol consisting of 900 pulses on the firing rate and structural remodeling [i.e., connectivity between stimulated neurons (S–S), between non-stimulated excitatory neurons (E–E), and between stimulated and non-stimulated neurons (S–E and E–S)]. (D) Effects of the same stimulation protocol on the firing rate of stimulated neurons and connectivity between stimulated neurons at the four representative amplitudes from Fig 2C [i.e., weak (a), strong-equivalent (b), peak (c), and strong (d)].

end of stimulation, which was also marked by a sudden drop in the net input received by the non-stimulated excitatory and inhibitory neurons, led to an instant drop in firing rates. This was followed by the formation of new connections that compensated for the now reduced network activity. As activity returned to baseline, a reorganization of network connectivity became evident: The stimulated neurons showed significantly more connections among each other (S-S), while the connection between the stimulated and non-stimulated neurons (S-E) was reduced; the connectivity among the non-stimulated neurons (E-E) remained unaltered. These simulations suggest, that rTMS-like electric stimulation can have distinct effects on the connectivity among and between stimulated and non-stimulated neurons, as reported before (c.f., Fig 2 of [36]).

## Dose-dependent effects of rTMS on structural network remodeling

We also assessed the outcome of the different stimulation intensities on homeostatic structural plasticity and network connectivity (Fig 3D). The same stimulation protocol (10 Hz, 900 pulses) was applied with weak, peak, strong, and strong-equivalent intensities (c.f., Fig 2B and 2C). As shown in Fig 3D, the largest change in the connectivity among the stimulated neurons was seen in response to the peak amplitude (i.e., a 68 mV membrane potential increase). The weak amplitude elicited a small response in neural activity, and only minor changes in lasting connectivity were observed (Fig 3D). The strong and strong-equivalent amplitudes yielded different effects on connectivity. The network receiving strong-amplitude stimulation failed to rapidly restore its activity to baseline by homeostatic structural plasticity during stimulation, which was reflected in a weaker overall connectivity change. This may be attributed to the recurrent inhibition recruited by a strong electric stimulation, which then affected the stimulated neurons. This phenomenon was not observed in the strong-equivalent stimulation, while a considerable remodeling of network connectivity was noted (Fig 3D).

## Influence of the stimulation duration on network remodeling

We observed that the changes in network connectivity following stimulation were directly proportional to the degree of reorganization induced during the stimulation process (Fig 4A). This observation had important implications for the stimulation duration, including the number of pulses applied at a given frequency. The finding suggests that once the firing rate of stimulated neurons is restored via network reorganization during stimulation, the application of additional pulses will not have a further effect on the outcome of the intervention, at least not in terms of lasting changes in network connectivity after stimulation.

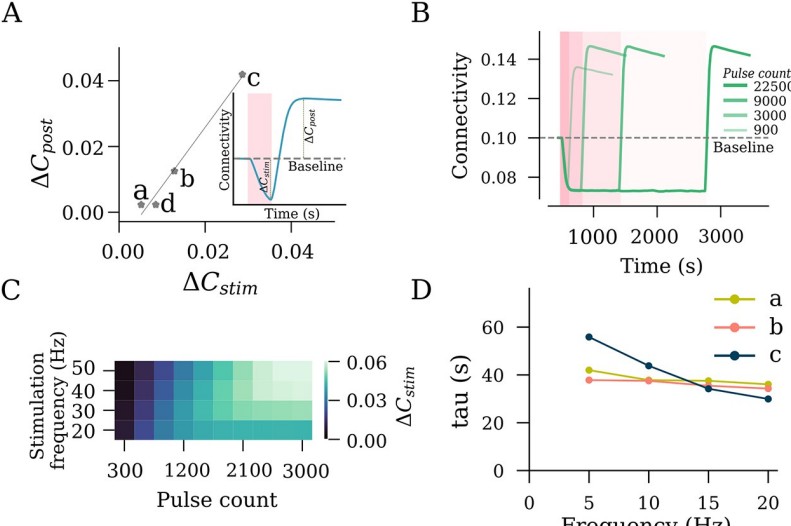

**Fig 4. rTMS intensity and pulse number affect the structural remodeling of the stimulated population (S).** (A) Interrelation between the $S-S$ connectivity drop during stimulation ($\Delta C_{stim}$) and $S-S$ connectivity increase post stimulation ($\Delta C_{post}$). (B) $S-S$ connectivity changes from different pulse numbers of 10 Hz stimulation at peak stimulation intensity (c, as defined in Fig 2C). (C) Saturation points of $S-S$ connectivity, expressed in the form of total pulse numbers required to reach saturation, are summarized for a range of frequencies. (D) Time constants of connectivity decay ($\tau_{decay}$) were extracted by fitting an exponential function to connectivity drop among stimulated neurons (S–S).

To explore this hypothesis, we applied 10 Hz stimulations of different durations to 10% of the excitatory neurons and assessed the trajectories of connectivity among the stimulated neurons (Fig 4B). We observed an increasing post-stimulation peak connectivity with an increasing stimulation duration. However, this relationship did not hold beyond a certain point. For 10 Hz stimulation, we found that stimulation beyond ∼3000 pulses did not contribute to further changes in the peak connectivity. This allowed us to conclude that the connectivity change has reached a saturation point, and 10 Hz stimulation for longer durations would not have a stronger effect on network connectivity (Fig 4B). Indeed, the outcome of a stimulation with 22500 pulses was comparable to that observed with 3000 and 9000 pulses, as shown in Fig 4B.

We followed up on this observation by extending our simulations to include a range of frequencies from 10 Hz to 50 Hz, as summarized in Fig 4C. The trend of connectivity saturation was maintained, with lower frequencies taking larger pulse numbers to reach the saturation point. Considering that the pulse number is a product of stimulation duration and frequency, it is useful to evaluate the impact of stimulation duration on connectivity saturation as well. For this, we extracted the time constant of decay ($\tau_{\text{decay}}$) by fitting exponential curves to average connectivity between stimulated neurons during stimulation (Fig 4D). The $\tau_{\text{decay}}$ values across different frequencies at a fixed stimulation intensity were comparable, with a trend of inverse proportionality in case of peak stimulation intensity. We deduce that the total stimulation duration has a major impact on the net stimulation outcome, irrespective of the frequency.

## Effects of the clinically approved iTBS protocol on network activity and connectivity

Finally, we evaluated the effects of the clinically approved iTBS protocol, which has a more complex stimulation pattern with inter-train intervals (Fig 5A). We systematically applied the four relevant stimulation intensities, namely weak, peak, strong, and strong-equivalent, and assessed the changes in network connectivity (Fig 5B). Similar to what we observed with 10 Hz stimulation, the weak and peak stimulation intensities led to small and large changes in connectivity, respectively. Comparatively, the strong-equivalent intensity induced intermediate changes in connectivity, while the strong stimulation intensity led to only small changes in connectivity.

We then evaluated different stimulation durations, including the pulse numbers at peak stimulation intensity, and found that a plateau was reached between 600 and 1200 pulses, with 900 pulses showing approximately the same effect as 1200 pulses on network connectivity (Fig 5C). An additional increase in connectivity was evident at 1500 pulses, indicating that unlike the 10 Hz stimulation protocol, the iTBS protocol may assert additional effects when large numbers of pulses are applied. Indeed, the simulations with 3000 and 9000 pulses (c.f., Fig 4B) confirmed this suggestion (Fig 5D). Notably, the effects of the iTBS protocol on structural remodeling were weaker than those of the pulse-matched 10 Hz stimulation protocol (Fig 5D). This difference may be attributed to the inter-train interval of the iTBS protocol. Consistent with this suggestion, pulse-matched continuous TBS (cTBS) induced structural remodeling that exceeded the effects of iTBS and 10 Hz stimulation (Fig 5D). Taken together, these results emphasize the relevance of proper selection of stimulation parameters, specifically the stimulation intensity and pulse number, where "overdosing" may have negative or at least no additional desired effects.

## Discussion

In this study, we explored the effects of rTMS on network dynamics and connectivity using simulations of an inhibition-dominated recurrent neural network with homeostatic structural

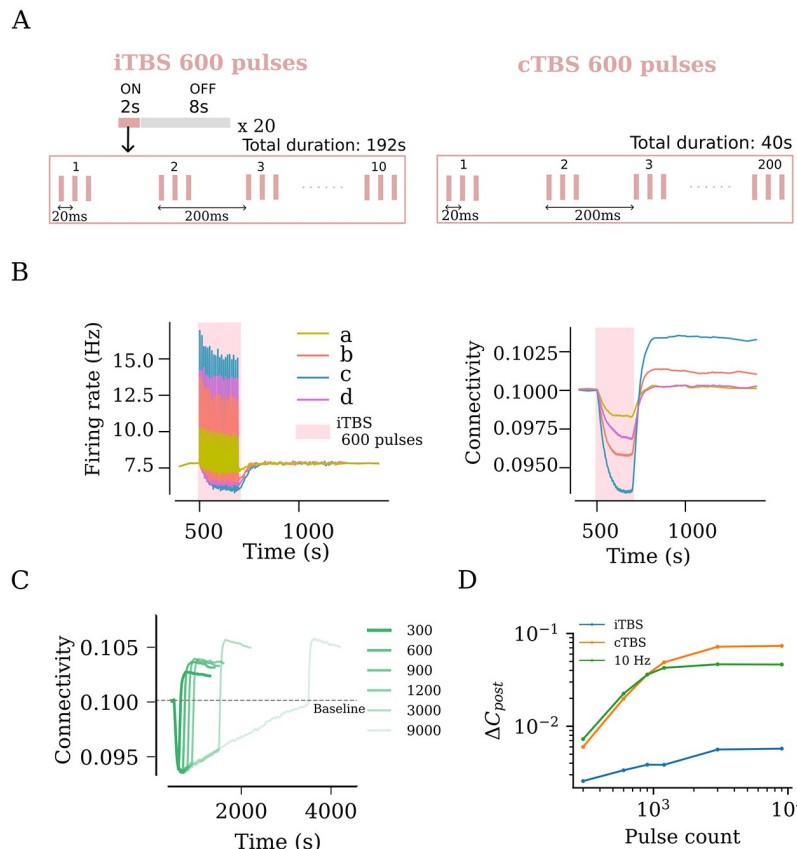

**Fig 5. rTMS leads to duration and intensity dependant overstimulation for intermittent Theta Burst Stimulation (iTBS).** (A) US FDA approved iTBS protocol consists of 600 pulses distributed across ON times of 2 s and OFF times of 8 s. The ON times consist of ten bursts of stimulus pulses at 5 Hz, where each burst consists of 3 pulses occurring at 50 Hz. (B) iTBS applied at peak amplitude (c, as defined in Fig 2C) resulted in the strongest firing rate response and the largest network connectivity upshoot. (C) iTBS at increasing stimulation duration (i.e., pulse numbers) was found to cause increasing values of post-stimulation connectivity upshoot among stimulated neurons. This trend was tested for iTBS, cTBS and 10 Hz and is summarised as log-log plots in (D).

plasticity. rTMS was found to increase the activity of neurons and induce characteristic changes in network connectivity. These effects of rTMS depended on the stimulation intensity, frequency, and duration. Differential effects of rTMS were observed in the stimulated and non-stimulated neurons; the connectivity among the stimulated neurons increased, while disconnection between the stimulated and non-stimulated neurons was observed. Our simulations suggest that recurrent inhibition, which is recruited at high stimulation intensities, may counter rTMS-induced neural activation and plasticity. We also observed that increasing the number of stimulation pulses beyond a certain point may saturate the structural network reorganization. Thus, stimulation protocols may exist, where no additional desired effects will be observed by further increasing the intensity of stimulation or number of TMS pulses. However, for the FDA-approved iTBS protocol, we observed an additive effect on the changes in network activity at larger pulse numbers. We attribute this effect to the complex pattern of the iTBS protocol, specifically the inter-train intervals. iTBS at 900 pulses seems to be more effective than iTBS at 600 pulses in our simulations. Notably, however, the effects of iTBS on the structural remodeling of the stimulated networks were weaker than those of pulse-matched 10 Hz stimulation or cTBS. Taken together, our results suggest a new mechanism of rTMS-

induced plasticity that does not depend on LTP-like plasticity and synaptic weight changes. This rTMS-induced homeostatic structural plasticity is sensitive to specific parameters of the stimulation protocol, emphasizing the need for a careful standardization and a systematic experimental assessment of dose-response relationships in rTMS-based basic and clinical studies.

Although direct experimental evidence on the human neocortex is still lacking, it seems well established in the field that rTMS changes cortical excitability by modulating excitatory and inhibitory neurotransmissions [27, 65, 66]. However, the effects of rTMS on cortical excitability—measured as changes in the amplitudes of motor evoked potentials—return to baseline within 90 min after stimulation. Therefore, it is unlikely that rTMS-induced LTP or long-term-depression (LTD) is the major or sole mechanism underlying the therapeutic effects of rTMS that can last weeks or months after stimulation [67, 68]. Yet, clinical protocol designs are often based on studies reporting rTMS-induced LTP- or LTD-like plasticity [7, 69, 70]. Herein, we used computational modeling to explore an alternative biomechanism of rTMS that is based on homeostatic plasticity and structural remodeling of neuronal networks, namely homeostatic structural plasticity. Homeostatic plasticity involves activity-dependent negative-feedback mechanisms that aim at maintaining neuronal networks within a stable operational range [71–73]: An increase in network activity leads to weakening of excitatory synapses, strengthening of inhibitory synapses, and therefore shifting in the excitability of neurons. Previously, Gallinaro and Rotter [35] demonstrated emergent associative properties of homeostatic structural plasticity, via activity-driven formation of neuronal ensembles. A similar modeling approach has also been used to explore effects of transcranial Direct Current Stimulation [36]. Consistent with these previous findings and with the use of a similar computational approach, the present results suggest that rTMS triggers an activity-dependent disconnection of neurons that enables the formation of new excitatory synapses and leads to a profound structural remodeling of stimulated networks.

While some experimental evidence supports the existence of homeostatic structural plasticity [74–76], for a review article, see [77], its biological significance and the underlying molecular mechanisms warrant further investigation. In our previous investigation, utilizing live cell microscopy to examine the impact of repetitive transcranial magnetic stimulation (rTMS) on dendritic spines of cultured hippocampal CA1 neurons, we did not observe any significant changes in synapse numbers, including alterations in spine density, following 10 Hz stimulation [12]. This is consistent with the finding of a recent *in vivo* two-photon imaging study demonstrating subtle structural changes in dendritic spines in response to repeated sessions of low-intensity rTMS [14]. Notably, studies conducted by Ma et al. [29, 30] have also shown effects of rTMS on structural plasticity in mice/rats. Homeostatic structural plasticity could potentially involve the (un)silencing of synapses as a biological mechanism [77–79]. Synapses present on small dendritic spines or filopodia, primarily containing NMDA receptors, are often considered "silent" due to the blockage of NMDA receptors by magnesium ions at the resting membrane potential. However, these synapses can be activated upon the accumulation of depolarizing AMPA receptors [80–83]. Interestingly, our previous research unveiled that 10 Hz repetitive magnetic stimulation promotes the accumulation of AMPA receptors at existing small spine synapses, triggering the growth of these dendritic spines, which are presumably silent [12, 13]. Consequently, rTMS may facilitate homeostatic structural plasticity by enabling neurons to establish or eliminate functional synaptic connections through the regulation of AMPA receptor accumulation at preexisting synapses, without necessitating the recruitment of the complete molecular machinery for the formation or removal of new spines or synapses (c.f., [79]). While experimental evidence exists supporting the presence of homeostatic

structural plasticity [84–86], additional research is required to unravel its implementation time scale, biological significance, and the underlying molecular mechanisms.

In a network without structural plasticity, we observed a non-linear relationship between the stimulation intensity and neuronal firing rate changes. This non-linearity in the firing rate response can be attributed to recurrent inhibition. We observed increasing feedback inhibition in response to higher stimulation intensities. This effect had a major impact on the outcome of rTMS-induced structural plasticity. Accordingly, we defined four critical stimulation intensities for closer examination: weak, peak, strong and strong-equivalent. At amplitudes below the peak value, the inhibitory subpopulation was not strongly activated. Meanwhile, with stimulation stronger than the peak amplitude, stronger recurrent activity recruited the inhibitory subpopulation, which consequently inhibited the stimulated subpopulation, causing a weaker firing rate response. Indeed, stimulation stronger than the peak amplitudes yielded weaker effects on structural remodeling than did stimulation at a lower intensity, despite their comparable effects on the firing rates of the stimulated neurons. In general, this highlights the important role of inhibitory networks in rTMS-induced plasticity. Experimental evidence suggests that single-pulse TMS exerts inhibitory effects on neocortical dendrites by directly activating axons in the upper cortical layers, subsequently leading to the activation of dendrite-targeting inhibitory neurons within the neocortex of mice [87]. Our previous experimental research has further demonstrated that 10 Hz rTMS induces remodeling of inhibitory synapses [16]. Specifically, we observed reductions in dendritic inhibition, as well as changes in the strength, sizes, and numbers of inhibitory synapses onto pyramidal neurons following stimulation. These findings underscore the structural modifications occurring in inhibitory networks as a result of rTMS. Notably, rTMS has also been shown to trigger the remodeling of visual cortical maps [88, 89]. However, the direct effects of stimulation on inhibitory neurons and the role of homeostatic structural plasticity in inhibitory synapses remain elusive. To gain a deeper understanding of these dynamics, further investigation is needed to explore the dose-dependent effects on specific inhibitory neuron types and their implications for rTMS-induced structural remodeling of excitatory and inhibitory synapses [90–92]. These considerations highlight the importance of investigating the direct impact of stimulation on inhibitory neurons and elucidating the interplay between inhibitory synaptic plasticity and homeostatic structural plasticity. Importantly, our findings suggest that stronger stimulation may result in less effective structural remodeling of stimulated networks compared to weaker stimulation that produces equivalent changes in firing rates. This highlights the complexity of the relationship between stimulation parameters, network dynamics, and structural plasticity. Further research is warranted to fully unravel these relationships and optimize stimulation protocols for achieving desired structural remodeling outcomes.

Our model also makes predictions relevant for translational applications of rTMS. Based on our findings, we propose a model of "connectivity saturation". Stimulating networks of neurons may initiate homeostatic synaptic remodeling that leads to loss in connectivity among the neurons. The end of stimulation period is followed by further synaptic remodeling causing increase in connectivity among the stimulated neurons. We used an exponential function to fit the trajectory of connectivity during the stimulation period and extracted time constants of connectivity decay, $\tau_{\text{decay}}$. This value can be roughly interpreted as the least time required to attain structural equilibrium during stimulation. This translates to the maximum remodeling that is attainable once stimulation is turned off. We found that the $\tau_{\text{decay}}$ values were comparable for low stimulation intensities across a wide range of frequencies, emphasizing the relevance of the stimulation duration rather than the pulse numbers. At the peak stimulation intensity, we found a slight frequency dependency indicating, that lower frequencies take a longer time to achieve connectivity saturation. A similar connectivity saturation was not

observed in the iTBS protocol. However, the effects of iTBS on structural remodeling were much weaker than those of pulse matched 10 Hz stimulation or cTBS. This effect may be attributed to the inter-train intervals, which enabled the network to rewire during the stimulation protocol. In this context, it is crucial to acknowledge that the employed point neuron model falls short in capturing the intricate biophysical characteristics, neuronal diversity, and complex cyto-/fiberarchitecture observed in the brain. Additionally, it fails to accurately represent the various subtypes of neurons, including specific subtypes of principal neurons and interneurons, and their corresponding connectivity patterns. Moreover, the model did not attempt to capture the structural complexities of dendritic and axonal morphologies, leading to limitations in describing cell-type and input-specific effects. As a consequence, the model necessitated relatively high stimulation intensities to elicit action potentials, making it challenging to directly translate these absolute values into realistic effects of single-pulse TMS on individual neurons and small networks. To overcome these limitations and achieve accurate predictions, it is possible to employ multi-scale modeling approaches that incorporate biophysically realistic neurons and account for physiological network activity [61, 62]. Nevertheless, it is important to note that we provided ab initio thinking on how external stimulation should interfere with network dynamics and structural plasticity without being masked by the heterogeneous neural morphology, which could serve as the building blocks in the future to systematically understand the impact of neural morphology in TMS effects. To confirm and expand upon the relevant predictions obtained in our computer simulations, it is essential to develop translational frameworks that integrate computational models with *in vitro* and *in vivo* animal studies, as well as experiments conducted in healthy individuals. These combined approaches will enable the validation and extension of our findings on dose-response relationships. In addition, future models should incorporate rTMS-induced synaptic plasticity, encompassing changes in synaptic weights for both excitatory and inhibitory synapses, while considering its interplay with homeostatic structural plasticity (c.f., Kromer and Tass [93] who conducted a systematic assessment of stimulation parameters in networks of leaky integrate-and-fire neurons with spike-timing-dependent plasticity or Lu et al., [94] who explore the intricate interplay of homeostatic synaptic scaling and structural plasticity). We have confidence that these computational models will play a crucial role in evaluating the effects of rTMS under disease conditions, thereby informing protocol designs in clinical practice (c.f., Manos et al., [95]). Currently, in the field of TMS these designs heavily depend on studies that report rTMS-induced LTP or LTD-like plasticity in the motor cortex and do not effectively consider the intricate dynamics and states of complex neural networks and their relationship with structural plasticity. By incorporating these computational models, we can better understand and optimize the effects of rTMS, paving the way for more effective therapeutic interventions.

## Supporting information

**S1 Fig. rTMS intensity and pulse number affect the structural remodeling between stimulated (S) and non-stimulated excitatory (E) populations.** (A) Interrelation between the $S - E$ connectivity drop during stimulation ($\Delta C_{stim}$) and $S - E$ connectivity increase post stimulation ($\Delta C_{post}$). (B) $S - E$ connectivity changes from different pulse numbers of 10 Hz stimulation at peak stimulation intensity (c, as defined in Fig 2C). (C) Saturation points of $S - E$ connectivity, expressed in the form of total pulse numbers required to reach saturation, are summarized for a range of frequencies.
(TIF)

**S2 Fig. rTMS intensity and pulse number affect the structural remodeling between non-stimulated excitatory (E) neurons.** (A) Interrelation between the $E - E$ connectivity drop

during stimulation ($\Delta C_{stim}$) and $E − E$ connectivity increase post stimulation ($\Delta C_{post}$). (B) $E −$ $E$ connectivity changes from different pulse numbers of 10 Hz stimulation at peak stimulation intensity (c, as defined in Fig 2C). (C) Saturation points of $E − E$ connectivity, expressed in the form of total pulse numbers required to reach saturation, are summarized for a range of frequencies.
(TIF)

## Acknowledgments

We thank Dr. Júlia V. Gallinaro for valuable discussions. We thank Dr. Sandra Diaz-Pier from the Forschungszentrum Jülich for support on features of NEST. We thank the Freiburg University Computing Center for computational resources and support that contributed to the results of this study.

## Author Contributions

**Conceptualization:** Stefan Rotter, Andreas Vlachos.

**Formal analysis:** Swathi Anil.

**Funding acquisition:** Andreas Vlachos.

**Investigation:** Swathi Anil, Han Lu.

**Methodology:** Swathi Anil, Han Lu, Stefan Rotter.

**Project administration:** Andreas Vlachos.

**Resources:** Andreas Vlachos.

**Supervision:** Stefan Rotter, Andreas Vlachos.

**Visualization:** Swathi Anil.

**Writing – original draft:** Swathi Anil, Andreas Vlachos.

**Writing – review & editing:** Swathi Anil, Han Lu, Stefan Rotter, Andreas Vlachos.

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
