## [Decision Letter · Decision Letter 0]

28 Apr 2023

Dear Prof. Dr. Vlachos,

Thank you very much for submitting your manuscript "Repetitive transcranial magnetic stimulation (rTMS) triggers dose-dependent homeostatic rewiring in recurrent neuronal networks" for consideration at PLOS Computational Biology.

As with all papers reviewed by the journal, your manuscript was reviewed by members of the editorial board and by several independent reviewers. In light of the reviews (below this email), we would like to invite the resubmission of a significantly-revised version that takes into account the reviewers' comments. We would like to direct the authors' attention particularly to the major comments of the reviewer 3 - it is crucial to resolve the issues raised there.

We cannot make any decision about publication until we have seen the revised manuscript and your response to the reviewers' comments. Your revised manuscript is also likely to be sent to reviewers for further evaluation.

Sincerely,

Boris S. Gutkin

Academic Editor

PLOS Computational Biology

Marieke van Vugt

Section Editor

PLOS Computational Biology

Reviewer's Responses to Questions

**Comments to the Authors:**

Reviewer #1: In this manuscript the authors study effects of repetitive transcranial magnetic stimulation (rTMS) on recurrent networks with homeostatic plasticity. The goal in itself is interesting and worthwhile. The major problem however I have with the manuscript is that the authors use leaky integrate and fire model without any spatial dimension. Is this model at all capturing effects of magnetic stimulation, what about dendritic effects? I understand that things get get quickly complicated but at least limitations of this study should be discussed.

Could these results be compared to some experimental ones, like the one referenced in 12? or dissociated cell cultures?

Reviewer #2: In their manuscript, “Repetitive transcranial magnetic stimulation (rTMS) triggers dose-dependent homeostatic rewiring in recurrent neuronal networks,” the authors study the effect of stimulation on neuronal networks with and without homeostatic plasticity computationally. The authors find that stimulation of a neuronal subpopulation shapes the firing rate of different neuronal populations in a nonlinear manner. The maximum firing rate is reached at intermediate stimulation amplitudes as stronger stimulation leads to the recruitment of inhibitory neurons, which then reduces the overall firing rate. Stimulation-induced firing rate changes trigger homeostatic plasticity and cause a rewiring of the network. It may also lead to long-lasting changes in the network structure, as quantified by a change of the connection probability.

The paper is very interesting and contains several new results. Since the work is motivated by TMS, it is certainly of interest to the readership of PLoS CB. Furthermore, the paper is well-written. However, there are some major points that the authors need to address before I can recommend publication in PLoS CB. Below I give a detailed description of these major points and some minor changes.

Major points

1) The computational network model is not well motivated. Since the authors mean to study TMS, it would be good to state in detail which cortical region is considered and to which extent the network parameters are motivated by experimental data. Of particular importance would be to see whether the firing rates of excitatory and inhibitory neurons and the time scale on which the homeostatic plasticity remodels the network are chosen according to experimental data. I think the time scale of remodeling seems relatively fast (Table 3). What are the time scales on which synaptic remodeling occurs in the considered brain region? Furthermore, are the suggested stimulation amplitudes realistic? It seems that very strong stimulation would be required to achieve the suggested changes of the membrane potential.

2) The authors mention that they study an inhibition-dominated network; however, balanced networks are usually studied in the field. Maybe I am missing something here, but why is the network inhibition-dominated? In Figure 5, it seems that the firing rate before stimulation is constant, so isn’t the network in a balanced state before stimulation, where excitation and inhibition balance each other? Please explain the term inhibition-dominated in more detail and discuss how this affects the results.

3) Although the authors model rTMS, the stimuli are modeled as rectangular current pulses. There are numerous papers studying the effect of electrical stimulation in neuronal networks with STDP, some of which find similar effects of stimulation, i.e., synaptic reshaping of the synapses between stimulated neurons, e.g., Lubenov et al. Neuron 58, 118 (2008) and Kromer and Tass PLoS Comput. Biol. 18, e1010568 (2022), synapses between stimulated and not stimulated neurons Morrison et al. Neural Comput.19, 1437 (2007). The difference seems to be that for STDP the synaptic reshaping depends on the timing between spikes and the transmission delays, whereas the presented results suggest that it mainly depends on the differences in firing rates in networks with homeostatic plasticity. Note that STDP also leads to firing rate-dependent changes of the mean synaptic weights (see Burkitt et al. Neural Comput. 16, 885 (2004)). Are there effects that can be observed in networks with homeostatic plasticity that cannot be observed in networks with STDP? Please compare the observed effects to the results of studies on similar stimulation in networks with STDP. After all, stimulation would likely trigger multiple types of plasticity. In particular, in Kromer and Tass PLoS Comput. Biol. 18, e1010568 (2022), the impact of the stimulation parameters, e.g., the pulse number, the stimulation frequency, and the stimulation pattern (e.g., bursting, periodic) was analyzed in detail in networks of LIF neurons with STDP. Are the presented results somewhat comparable to their results? Please discuss.

4) Panel D of Figure 3 indicates that the studied network is multistable and that TMS can induce transitions between the different stable states. This is reminiscent of works on coordinated reset stimulation, where the goal was to induce a transition from a pathological state to a physiological (Manos et al. Frontiers in Physiology 12, 716556 (2021)). Based on the presented results, would TMS be able to induce such a transition? Please discuss.

5) Equation 3 is unclear to me. In line 109 z_i^k, was introduced as a “discrete number,” however, in Equation 3, it seems to be a (positive) real number. How are z_i^k and the numbers of dendritic spines and axonal boutons related? Furthermore, I guess k should be either “a” or “d” (lines 109 and 110) and not “pre” or “post” (Equation 3). Please clarify the notation and the definition of z_i^k.

6) In Figure 4, the overall drop in connectivity is studied. I wonder whether this is the best quantity to study. Wouldn’t it yield more insight if one would distinguish between the connection probability between the stimulated subpopulation and the other subpopulations? Please elaborate.

Minor

Throughout the paper: Please increase the font size in the figures. In some cases, it is very difficult to read the labels in the figures.

Line 57: “spike trains” -> “spike train”

Line 67: Why is the reset potential above the resting potential? This does not seem to be the case in Fig. 1D.

Line 80: What is the motivation for the asynchronous-irregular resting state? Is this motivated by a certain cortical region? What other states exist in the network?

Table 2: Please distinguish between static and plastic network.

Line 87: “equilibrium status” I guess this should be “equilibrium state”.

Line 110: “axonal buotons” -> “axonal boutons”

Line 122: “by amount, beta…” -> “by beta…”

Line 145: “modeled after output of conventional rTMS devices” Please double-check whether this sentence is correct.

Lines 150, 151: “cite”-> “site”

Line 160: “The effect of rTMS over networks” I guess this should be “The effect of rTMS on networks” instead.

Line 169: “protocol structure” This term is unclear to me. Please explain what this term refers to.

Line 171: FDA-approved for what?

Line 175: “rTMS pulse triggering membrane potential deviation” I guess this should be “rTMS pulse-triggered membrane potential deviation” instead.

Line 184: “nonidentical but equal net Poisson inputs” This term is unclear to me. Are you talking about different realizations of the Poisson inputs here?

Line 192 and later in the manuscript: “followed by an 8s interval” Interval of what? I guess the authors mean an 8s pause or an 8 second inter-burst interval. Please clarify throughout the manuscript.

Throughout the manuscript: “integrate and fire” -> “integrate-and-fire”

The discussion of the Ohms’ Law on page 7 is a little confusing. How are U,A, and R related to the integrate-and-fire model in Equation 1?

Line 205: “formulation” I guess this should be “equation” instead.

Line 210: “across defined time-steps” This is unclear to me. Is the firing rate based on the spike count in a 1000s interval?

Line 211: “from the group”->”in this population”

Line 218: “were” -> “was”

Line 219: “mean of the whole matrix” I would write “mean number of synapses between two neurons”.

Line 239: “repeated” -> “repetitive”?

Line 242: “experimental setting” Please write “simulation” instead. The authors did not perform experiments.

Figure 2: Please clarify whether homeostatic plasticity is ON or OFF in this network. Please also clarify for which population of neurons the firing rate is shown in Figure E.

Throughout the paper: Please write “different stimulation protocols” instead of “distinct stimulation protocols”

Line 345: It is unclear which “two parameters” the authors are talking about here.

Lines 347-350: Please specify which neuronal population is considered here.

Figure 3B: It is unclear to me whether the firing rate saturates here. Also, please write “Poisson input” instead of “Poissonian input”.

Lines 364-365: Please rewrite. What does the first part of the sentence have to do with the second part? It is unclear which “connectivity” is studied here? Is this the connection proability? Please clarify.

Line 408-410: Optimality with respect to what? Is the goal here just to increase the firing rate of the stimulated neuronal population? Please clarify.

Line 445: “overview”->”a review article”

Line 463: “need to recruit” -> “need for recruiting”?

Line 495: Please write “may initiate homeostatic synaptic remodeling”.

Line 498: What are the “affected neurons”? Are this the stimulated neurons?

Reviewer #3: Based on previous computational modeling studies, the authors introduce a model to describe the effect of rTMS-like stimulation on the reorganization of cortical connectivity. This approach is interesting and modeling studies could indeed help to better understand rTMS. However, in the current state of the manuscript, the link to rTMS remains superficial. Furthermore, although the idea of structural plasticity is very appealing, the authors miss to show how it can be distinguished from the state-of-the-art LTP/LTD-hypothesis. Unfortunately, also some model assumptions stay unclear.

Major Points:

- Cortical circuits show many different types of activity-dependent plasticity mechanisms and it is unclear which mechanisms are triggered by rTMS. With this model at hand, I am wondering how the network changes induced by structural plasticity differ from changes induced by LTP/LTD or other mechanisms. Such investigations could also lead to specific hypotheses that could be tested to verify whether structural plasticity is the main process driving TMS-dependent network reorganization. But for this, also the next point has to be clarified.

- What is the clinical/functional meaning of the observed connectivity remodeling? Is more remodeling better or worse? Or is a specific type of remodeling being needed? At the moment, it remains unclear how to relate the results from the model to results from rTMS-studies.

- Eq. 3: The authors are considering the same rule for the growth of axonal and dendritic elements. However, in literature, several models consider that axonal elements behave different to dendritic ones. How critical is this assumption of same rules?

- Throughout the manuscript the notation is sometimes confusing. In Eq. 1 the impact of rTMS is given by \\Delta V_rTMS. In Table 4 is written \\Delta V and in the results section is written \\Delta V_m. Are all these parameters the same?

- Figure 1B,C: I am confused here. In Eq. 1, the authors introduce that rTMS stimulation induces directly a change in membrane potential. However, here they introduce a current I(t). What is the relation? How is panel C linked to panel B?

- Why should a rTMS stimulation depolarize only excitatory neurons? I would also expect that some inhibitory neurons are affected. Would this change the results?

Minor Points:

- Line 67: The reset potential is noted by V_reset and in table 1 by V_r. Also, why is the reset potential higher than the resting potential? Often set it below the resting potential to emulate the refractory period.

- How does r_ext goes into Eq. 1?

- Lines 153-168: These paragraphs should be moved to the discussion section.

- Fig. 1D: How can the membrane potential without stimulation be at -60 mV if the resting potential is at 0mV? In addition, after a spike, in these plots the reset is even below -60 mV, although it is 10 mV.

- Fig. 2: What is the time axis in panel B? At which point of time after or during stimulation do you evaluate the firing rate to obtain panel C? a, b, c, and d are lines and not points in panel C.

**Have the authors made all data and (if applicable) computational code underlying the findings in their manuscript fully available?**

Reviewer #1: Yes

Reviewer #2: Yes

Reviewer #3: **No: **I did not find any statement or link to access the used code.

PLOS authors have the option to publish the peer review history of their article (what does this mean?). If published, this will include your full peer review and any attached files.

Reviewer #1: No

Reviewer #2: No

Reviewer #3: No
---

## [Decision Letter · Decision Letter 1]

11 Sep 2023

Dear Prof. Dr. Vlachos,

Thank you very much for submitting your manuscript "Repetitive transcranial magnetic stimulation (rTMS) triggers dose-dependent homeostatic rewiring in recurrent neuronal networks" for consideration at PLOS Computational Biology. As with all papers reviewed by the journal, your manuscript was reviewed by members of the editorial board and by several independent reviewers. The reviewers appreciated the attention to an important topic. Based on the reviews, we are likely to accept this manuscript for publication, providing that you modify the manuscript according to the review recommendations.

Sincerely,

Boris S. Gutkin

Academic Editor

PLOS Computational Biology

Marieke van Vugt

Section Editor

PLOS Computational Biology

Reviewer's Responses to Questions

**Comments to the Authors:**

Reviewer #1: The authors disused in depth the limitations of the model and compared their results more closely to experimental ones. I believe the manuscript is improved and can be now published.

Reviewer #2: The authors addressed most of my comments thoroughly. I only have a few further minor points, which I list below. Once these points have been addressed, the manuscript is suitable for publication. I do not need to see the manuscript again before publication.

Lines 45-46:

"This behavior contributes to neuronal intrinsic properties like stochastic firing, diverse input integration, absence of intrinsic oscillations, and sensitivity to input." It needs to be clarified what behavior the authors refer to. Reading this sentence, I would understand that "behavior" refers to the "asynchronous irregular neuronal activity," which would be a network property, and I would not consider a network property to contribute to the intrinsic properties of the neurons. My suggestion would be to write something along the lines of "The dynamics of individual neurons is characterized by stochastic firing, diverse input integration, absence of intrinsic oscillations, and sensitivity to input."

Line 114

The formulation "rendering our findings applicable to any neocortical region" in line 114 is a very strong statement that remains to be proven. Please reformulate.

Line 126

"These synapses are randomly established with a (10%)

connection probability." -> "These synapses are randomly established with a connection probability of 10%."

Reviewer #3: The authors have addressed all my concerns. Thank you very much.

**Have the authors made all data and (if applicable) computational code underlying the findings in their manuscript fully available?**

Reviewer #1: None

Reviewer #2: Yes

Reviewer #3: Yes

PLOS authors have the option to publish the peer review history of their article (what does this mean?). If published, this will include your full peer review and any attached files.

Reviewer #1: No

Reviewer #2: No

Reviewer #3: No

Figure Files:

Data Requirements:

Reproducibility:

References:

---

## [Editor Report · Decision Letter 2]

11 Oct 2023

Dear Prof. Dr. Vlachos,

We are pleased to inform you that your manuscript 'Repetitive transcranial magnetic stimulation (rTMS) triggers dose-dependent homeostatic rewiring in recurrent neuronal networks' has been provisionally accepted for publication in PLOS Computational Biology.

Best regards,

Boris S. Gutkin

Academic Editor

PLOS Computational Biology

Marieke van Vugt

Section Editor

PLOS Computational Biology

---

## [Editor Report · Acceptance letter]

9 Nov 2023

PCOMPBIOL-D-23-00418R2 

Repetitive transcranial magnetic stimulation (rTMS) triggers dose-dependent homeostatic rewiring in recurrent neuronal networks

Dear Dr Vlachos,

I am pleased to inform you that your manuscript has been formally accepted for publication in PLOS Computational Biology. Your manuscript is now with our production department and you will be notified of the publication date in due course.

With kind regards,

Judit Kozma
